# Immunomodulatory Effects of Radon Inhalation on Lipopolysaccharide-Induced Inflammation in Mice

**DOI:** 10.3390/ijerph191710632

**Published:** 2022-08-26

**Authors:** Takahiro Kataoka, Shota Naoe, Kaito Murakami, Yuki Fujimoto, Ryohei Yukimine, Ayumi Tanaka, Kiyonori Yamaoka

**Affiliations:** 1Faculty of Health Sciences, Okayama University, 5-1 Shikata-cho 2-chome, Kita-ku, Okayama-shi, Okayama 700-8558, Japan; 2Graduate School of Health Sciences, Okayama University, 5-1 Shikata-cho 2-chome, Kita-ku, Okayama-shi, Okayama 700-8558, Japan

**Keywords:** autoimmune diseases, cytokine, antioxidant function, lipopolysaccharide, radon inhalation

## Abstract

Typical indications for radon therapy include autoimmune diseases such as rheumatoid arthritis (RA). We had previously reported that radon inhalation inhibits Th17 immune responses in RA mice by activating Th1 and Th2 immune responses. However, there are no reports on how radon inhalation affects the activated Th1 and Th17 immune responses, and these findings may be useful for identifying new indications for radon therapy. Therefore, in this study, we investigated the effect of radon inhalation on the lipopolysaccharide (LPS)-induced inflammatory response, focusing on the expression of related cytokines and antioxidant function. Male BALB/c mice were exposed to 2000 Bq/m^3^ radon for one day. Immediately after radon inhalation, LPS was administered intraperitoneally at 1.0 mg/kg body weight for 4 h. LPS administration increased the levels of Th1- and Th17-prone cytokines, such as interleukin-2, tumor necrosis factor-α, and granulocyte-macrophage colony-stimulating factor, compared to no treatment control (sham). However, these effects were suppressed by radon inhalation. IL-10 levels were significantly increased by LPS administration, with or without radon inhalation, compared to sham. However, radon inhalation did not inhibit oxidative stress induced by LPS administration. These findings suggest that radon inhalation has immunomodulatory but not antioxidative functions in LPS-induced injury.

## 1. Introduction

Radon has been used to treat patients with autoimmune diseases, especially pain-related diseases such as rheumatoid arthritis (RA). Therapies using a hot spring with or without radon have been shown to be effective in treating patients with RA, wherein a hot spring with radon showed long-term efficacy compared to that without radon during the 6-month follow-up conducted by the study in [1]. Radon therapy also shows longer-lasting improvements in pain and results in reduced doses of medicines, such as corticosteroids and non-steroidal anti-inflammatory drugs [2]. A meta-analysis of radon therapy demonstrated a positive effect on pain in rheumatic diseases [3]. These studies suggest the clinical effectiveness of radon therapy in treating patients with RA.

Activation of antioxidative functions by radon inhalation plays a critical role in alleviating inflammation and pain [4]. Radon inhalation increases the amount of antioxidant substances such as superoxide dismutase (SOD), catalase (CAT), and total glutathione (t-GSH) in some organs of mice [5]. In addition, the liver and kidneys of normal mice have a high antioxidant function; the brain, pancreas, and stomach have a low antioxidant function and low lipid peroxide (LPO) levels; and the heart, lung, small intestine, and large intestine have low antioxidant function and high lipid oxide levels [6]. This results in the inhibition of several types of inflammation induced by reactive oxygen species (ROS), such as carrageenan-induced inflammatory paw edema [7], dextran sulfate sodium-induced colitis [8], and formalin-induced inflammatory pain [9]. Although some inflammatory cytokines were assayed in these previous studies, the balance among Th1, Th2, and Th17 cells remains to be elucidated.

Peripheral Th17/regulatory T cell (Treg) imbalance is involved in the development of RA. A report showed that the levels of Th17-related cytokines were higher, whereas the levels of Treg-related cytokines were significantly lower, in patients with RA than those in healthy individuals [10]. In our previous study, we reported differences in cytokine production between normal and RA mice following radon inhalation [11]. Radon inhalation for four weeks produced Th1-, Th2-, and Th17-prone cytokines in normal mice, whereas it produced only Th1- and Th2-prone cytokines and decreased Th17-prone cytokines in RA mice. As Th17 plays a critical role in the development of RA, the probable mechanism of radon therapy is the inhibition of the Th17 immune response. In addition, clarifying the effects of radon inhalation on different types of immune responses may be useful for identifying new indications for radon therapy. 

Lipopolysaccharide (LPS) is widely used to induce inflammation. LPS treatment activates the Th1 response, leading to the release of tumor necrosis factor (TNF)-α and interleukin (IL)-12 (p70) [12]. Furthermore, LPS directly stimulates Th17 differentiation in vitro [13]. However, the Th2 response is activated only by treatment with a low dose of LPS [14]. These studies indicate that LPS is a useful model for examining the effects of radon. However, no studies on the effects of radon inhalation on Th1 and Th17 cells following LPS treatment have been reported. Therefore, in this study, we aimed to examine the effects of radon on LPS-induced inflammatory responses, focusing on related cytokines and antioxidant function.

## 2. Materials and Methods

### 2.1. Animals

Eight-week-old male BALB/c mice were purchased from Jackson Laboratory (Yokohama, Japan). This experimental protocol was approved by the Animal Care and Use Committee of Okayama University.

### 2.2. Experimental Design

The mice were divided into three groups (n = 7, in each group): no treatment (sham, control), LPS administration after sham inhalation (LPS), and LPS administration after radon inhalation (Rn + LPS). The mice in the sham group received no treatment, and those in the LPS group received a sham inhalation followed by intraperitoneal administration of LPS at 1.0 mg/kg for 4 h. The mice in the Rn + LPS group inhaled radon at 2000 Bq/m^3^ for 24 h, followed by administration of LPS at 1.0 mg/kg. After 4 h of LPS administration, the mice were euthanized using CO_2_. The experiments were performed once. 

LPS administration induces oxidative stress [15], and radon inhalation activates antioxidative functions [5]. Therefore, we also studied the effects of radon therapy on the antioxidative functions of different organs of the mice by assessing the levels of LPO and other antioxidant enzymes. The dose and gap between LPS administration and euthanasia were determined based on our preliminary study, in which we tested the effects of different doses of LPS on inflammatory responses based on a previous study [15]. LPS administration at 0.4 mg/kg was sufficient to induce inflammatory responses (Figure A1). However, after 24 h of LPS administration, the levels of interferon (IFN) and IL-10 were almost equivalent to those in control. Considering these results and those of previous studies [15], we chose a dose of 1.0 mg/kg LPS for 4 h. Furthermore, in the preliminary study, the levels of IL-1β, -2, -4, -5, -6, -9, -10, -12 (p70), -13, -17, IFN-γ, granulocyte-macrophage colony-stimulating factor (GM-CSF), and TNF-α were assessed (Figure A1). However, the levels of IL-5, IL-9, IL-13, and IL-17 remained unchanged following radon inhalation or LPS administration. Therefore, we did not assess these cytokines in the present study.

### 2.3. Radon Inhalation

Radon inhalation was performed using a radon exposure system, which we have developed and reported in our previous study [7]. DOLL STONEs (Nigyotoge genshiryoku sangyo Co., Ltd. Tomata-gun, Okayama, Japan) were used as radon sources. The DOLL STONEs were set in a desiccator, and then radon in the desiccator was transferred to each mouse cage using an air pump. Radon concentration in the animal cage was maintained at approximately 2000 Bq/m^3^ during the experiment, and the mice were placed in this cage for one day to induce radon exposure. 

### 2.4. LPS Administration

Immediately after radon inhalation, LPS (LPS from *Escherichia coli* O127:B8 Sigma-Aldrich, Tokyo, Japan) at a dose of 1.0 mg/kg body weight was intraperitoneally administered once to the mice in the Rn + LPS group. Mice in the LPS group received the same dose, while those in the sham group did not receive LPS.

### 2.5. Sample Preparation

Blood was collected from the heart of all mice immediately after euthanasia using CO_2_ and centrifuged at 3000× *g* for 5 min at 4 °C. The upper aqueous layer was collected and preserved at −80 °C until use for cytokine assays. The brain, lungs, heart, liver, pancreas, kidneys, and small and large intestines were quickly removed without perfusion. Then, the organs were washed using distilled water before storing them at −80 °C until further analysis. 

### 2.6. Cytokine Assay

The levels of cytokines IL-1β, -2, -6, -10, -12 (p70), IFN-γ, and TNF-α were detected using the Bio-Plex Pro Mouse Cytokine Th1 Plex Panel kit (catalog no. L6000004C6; Bio-Rad, Hercules, CA, USA, Bio-Plex Pro Mouse Cytokine Th1 Plex Panel, 1 × 96-well, includes coupled magnetic beads, detection antibodies, standards, assay buffer, wash buffer, detection antibody diluent, streptavidin-PE, flat bottom plate, sealing tape, standard diluent, sample diluent, for the detection of IL-1β, IL-2, IL-6, IL-10, IL-12 (p70), IFN-γ, TFN-α). IL-4 and GM-CSF were determined using the Bio-Plex Pro Mouse Cytokine Singleplex Set kits (catalog no. 171G5005M and 171G5016M, respectively; Bio-Rad). The assay was consigned to Okayama University Hospital Biobank. The level of each cytokine was assessed following the instructions provided with the kits Data were acquired using a Bio-Plex 200 system (Bio-Rad). The limits of detection of IL-1β, IL-2, IL-4, IL-6, IL-10, IL-12 (p70), IFN-γ, GM-CSF, TNF-α, IL-5, IL-9, IL-13, and IL-17A were 9.4, 0.6, 2.1, 0.2, 1.0, 2.3, 1.2, 5.6, 1.4, 0.3, 12.5, 38.7, and 0.8 pg/mL, respectively.

### 2.7. Antioxidant Assay

The levels of SOD, CAT, t-GSH, LPO, and total protein in the organs were assessed using individual assay kits following previously described protocols [6,16]. For the SOD and glutathione assays, samples were homogenized on ice in a 10 mM phosphate buffered solution (PBS; pH 7.4), and homogenates were used for further analysis. SOD activity was determined using the SOD Assay Kit-WST (catalog no. S311; Dojindo Molecular Technologies, Inc., Kumamoto, Japan) based on the nitroblue tetrazolium (NBT) reduction method [17]. Briefly, the homogenates were centrifuged at 12,000× *g* for 45 min at 4 °C, and the supernatants were collected. SOD activity in the supernatants was measured by the extent of inhibition of NBT reduction measured at 450 nm using a spectrophotometer (Viento XS; DS Pharma Biomedical Co., Ltd., Osaka, Japan). Percentage inhibition of one-unit enzyme activity was defined as 50% inhibition of NBT reduction. 

CAT activity was measured using a Catalase Assay Kit (catalog no. PQ1; Cayman Chemical, Ann Arbor), which uses a method based on the reaction of the enzyme with methanol in the presence of an optimal concentration of H_2_O_2_. The formaldehyde produced was measured colorimetrically with 4-amino-3-hydrazino-5-mercapto-1,2,4-triazole (Purpald) as the chromogen. Purpald specifically forms a bicyclic heterocycle with aldehydes, which changes from colorless to a purple color upon oxidation [18,19]. The absorbance was read at 540 nm using a plate reader (Viento XS). 

The t-GSH content was measured using the Bioxytech GSH-420™ assay kit (catalog no. 707002; OXIS Health Products, Inc., Portland, OR, USA). This assay is based on the formation of a chromophoric thione, which is directly proportional to the t-GSH concentration and can be measured at 420 nm. Briefly, sample tissue homogenates were mixed with an ice-cold 10 mM PBS and 7.5% trichloroacetic acid solution. The homogenates were centrifuged at 3000× *g* for 10 min at 4 °C, and the supernatants were collected to measure t-GSH. 

LPO level was estimated using the Bioxytech LPO-586™ assay kit (catalog no. 21023; OXIS Health Products, Inc.). Briefly, the samples were placed in 10 mM PBS (pH 7.4), and 10 μL 0.5 M butylated hydroxytoluene in acetonitrile was added per mL of the buffer–tissue mixture. The tissue was homogenized, the homogenate was centrifuged at 15,000× *g* for 10 min at 4 °C, and the supernatant was used for the biochemical assay. The LPO assay is based on the reaction of the chromogenic reagent, N-methyl-2-phenylidole, with malondialdehyde, a product of lipid peroxidation, and 4-hydroxyalkenals at 45 °C. The optical density of the colored product was read at 586 nm using a spectrophotometer. The limitation of detection of LPO level was 0.1 nmol/mL, with an absorbance value of approximately 0.011. 

The protein content of each sample was measured by the Bradford method using a Protein Quantification Kit-Rapid (catalog no. 21012; Dojindo Molecular Technologies, Inc.) [20].

### 2.8. Statistical Analyses

Data are presented as the mean ± standard error of the mean. Statistical significance of differences was determined using a one-way analysis of variance followed by Tukey’s test for multiple comparisons. The data were considered statistically significant at *p* < 0.05.

## 3. Results

### 3.1. Effects of Radon Inhalation on LPS-Induced Differentiation-Inducing Cytokines in Serum

Regarding cytokines (IL-12 (p70), IL-4, and IL-6) that induce differentiation into helper T cells (Th1, Th2, and Th17 cells), LPS administration with or without radon inhalation significantly increased the levels of IL-12 (p70) and IL-6 compared to sham administration. However, radon inhalation significantly decreased IL-6 levels compared with LPS administration alone. IL-4 levels were not detected in any group (Figure 1). 

### 3.2. Effects of Radon Inhalation on LPS-Induced Th1-, Th17-, and Treg-Prone Cytokines in Serum

Next, the Th1-, Th17-, and Th2/Treg-prone cytokines were assayed. LPS administration, with or without radon inhalation, significantly increased the levels of IL-2 and IFN-γ, which are Th1-prone cytokines, and the levels of TNF-α and GM-CSF, which are Th17-prone cytokines. However, radon inhalation significantly decreased IL-2, TNF-α, and GM-CSF levels compared with LPS administration alone. IL-10 levels were significantly increased by LPS administration, with or without radon inhalation (Figure 2).

### 3.3. Effect of Radon Inhalation on Oxidative Stress Associated with LPS Administration in Various Organs

In livers with high antioxidant function, t-GSH content significantly decreased, and LPO level significantly increased, whereas SOD and CAT levels did not differ significantly, with or without radon inhalation. In kidneys with similar antioxidant function, the significantly reduced t-GSH content approached the levels of control after radon inhalation; however, no other parameters showed significant differences in any of the groups (Figure 3).

In the pancreas, which has a low antioxidant function and low LPO levels, the t-GSH content significantly increased with or without radon inhalation, whereas SOD and CAT levels did not differ significantly with or without radon inhalation. However, in the brain, the levels of none of the markers differ significantly among the three groups (Figure 4).

In lungs with low antioxidant function and high LPO levels, CAT activity significantly increased in the sham + LPS group compared to the sham group, and LPO levels significantly increased with or without radon inhalation compared to that of the no treatment control. However, the levels of LPO and CAT did not differ between sham + LPS and Rn + LPS groups (Figure 5). In large intestines with similar conditions, radon inhalation significantly suppressed the LPS-induced increase in SOD activity. No significant changes were observed in the levels of the oxidative stress markers among the three groups in hearts or small intestines with similar antioxidant functions (Figure 5). 

## 4. Discussion

Transforming growth factor-beta (TGF-β) is an anti-inflammatory cytokine that exerts immunomodulatory effects. Radon spa therapy increases TGF-β levels in the blood of patients with ankylosing spondylitis [21]. Radon balneotherapy alters the ratio between Th17 and Treg cells owing to an increase in Treg cells [22,23]. Our results showed that the level of IL-10, which is related to both Th2 and Treg cells, was significantly increased by LPS administration, with or without radon inhalation. The increased IL-10 level could be attributed to its anti-inflammatory nature, which suppresses the excessive immune response and maintains immune constancy. However, the level of IL-4, which is a Th2 cell differentiation induction cytokine, was not increased by LPS administration. Unlike Th1 and Th17 cells, which are involved in LPS-induced inflammation, Th2 cells are involved in allergic inflammatory responses, which explains the non-detection of IL-4 in this study. The results for Th1- and Th17-prone cytokines may indicate that radon inhalation regulates Th1 and Th17 cell responses via Treg cell activation. We had previously reported that radon inhalation inhibits Th17 cell responses related to RA development due to the activation of Th1 and Th2 cell responses [11]. Since the results of the present study show that LPS administration produces an immune response different from that observed in RA [11], we examined how radon inhalation affects the immune response induced by LPS administration. The results demonstrated that radon inhalation reduced LPS-induced inflammation by inhibiting the production of Th1- and Th17-prone cytokines. This is an important finding underlying the mechanisms and new indications for radon therapy.

LPS administration induces damage in the brain [24,25], lungs [26], heart [15], liver [27], pancreas [28], kidneys [29,30], and intestines [31]. Thus, oxidative stress and inflammation may play important roles in the development of LPS-induced damage. However, the dose of LPS used varies according to the animal model. In the present study, we determined the administration dose of LPS in a preliminary study (Figure A1). To further evaluate LPS-induced oxidative stress in each organ, we assayed antioxidative substances in the brain, lungs, heart, liver, pancreas, kidneys, small intestine, and large intestine. The LPO levels showed that LPS administration induced oxidative stress in the lungs and liver. Specifically, t-GSH levels were decreased in the liver, demonstrating that oxidative stress is much stronger in the liver than that in any other organ. The kidneys were also damaged by LPS administration, as observed by the decrease in t-GSH content. However, radon inhalation did not alleviate the LPS-induced oxidative stress under the present experimental conditions.

There may be clues to understanding the results of our previous studies. Although zymosan administration in SKG/Jcl mice (RA mice) increased IL-6 levels and decreased IL-12(p70) and IL-4 levels, radon inhalation decreased IL-6 levels and increased IL-12 (p70) and IL-4 levels. However, the antioxidative function in the lungs did not change [11]. In addition, the duration after LPS administration may be an important factor for evaluating oxidative stress in the organs. Our latest study suggests that although thoron inhalation, a radioisotope of radon, did not inhibit alcohol-induced liver damage at an early stage, it promoted early recovery from the damage [31]. For this reason, in the present study, we did not examine the time course effects of radon. Moreover, our preliminary study showed that the production of some cytokines induced by 0.4 mg/kg of LPS administration returned to the control levels 24 h after LPS administration (Figure A1).

## 5. Conclusions

This study, for the first time, demonstrated that Th1 and Th17 cells activated by LPS were inhibited by radon inhalation, indicating that radon inhalation exerts an immunomodulatory effect against LPS-induced damage. As radon inhalation activates antioxidative functions in mouse organs [5], the functions after radon inhalation are considered to regulate the immune response. Although our results showed that radon inhalation did not inhibit oxidative stress induced by LPS administration, the results of this study may be useful in identifying new indications for radon therapy. Radon inhalation conditions and the dose of LPS may affect our results regarding oxidative stress. In this study, we did not perform flow cytometry analyses. Therefore, further studies are required to gain insights into the precise source of cytokines produced under radon inhalation. Furthermore, future studies involving advanced techniques such as Western blotting or immunohistochemistry focusing on specific organs could further validate the alleviating effects of radon inhalation on LPS-induced inflammation.

## Figures and Tables

**Figure 1 ijerph-19-10632-f001:**
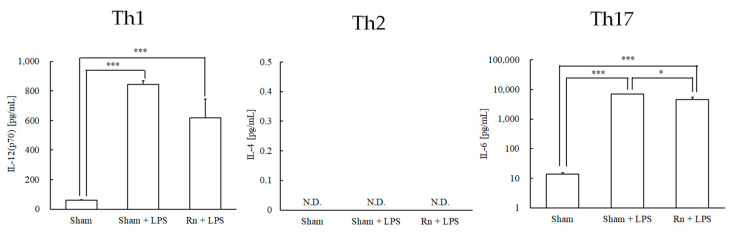
Effects of radon (Rn) inhalation on lipopolysaccharide (LPS)-induced T cell differentiation-inducing cytokines in serum. The results are presented as the mean ± standard error of the mean (SEM). N = 7; all experiments were conducted once; * *p* < 0.05, *** *p* < 0.001; IL, interleukin; N.D., not detected; sham, mouse group without any treatment; sham + LPS, mouse group that inhaled sham followed by LPS (1 mg/kg body weight); Rn + LPS, mouse group that inhaled radon (2000 Bq/m^3^) followed by LPS (1 mg/kg body weight).

**Figure 2 ijerph-19-10632-f002:**
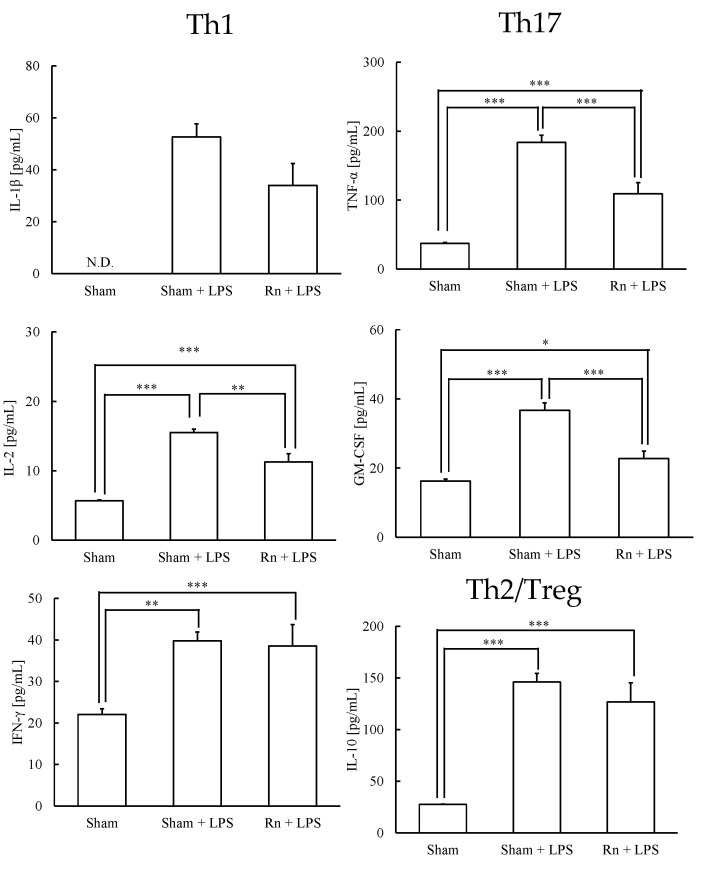
Effects of radon (Rn) inhalation on LPS-induced Th1-, Th17-, and Th2/Treg-prone cytokines in serum. The data show the mean ± SEM. N = 7; all experiments were conducted once. * *p* < 0.05, ** *p* < 0.01, *** *p* < 0.001; sham, mouse group without any treatment; sham + LPS, mouse group that inhaled sham followed by LPS (1 mg/kg body weight); Rn + LPS, mouse group that inhaled radon (2000 Bq/m^3^) followed by LPS (1 mg/kg body weight); IL, interleukin; TNF, tumor necrosis factor; GM-CSF; granulocyte–macrophage colony-stimulating factor; IFN, interferon; N.D., not detected.

**Figure 3 ijerph-19-10632-f003:**
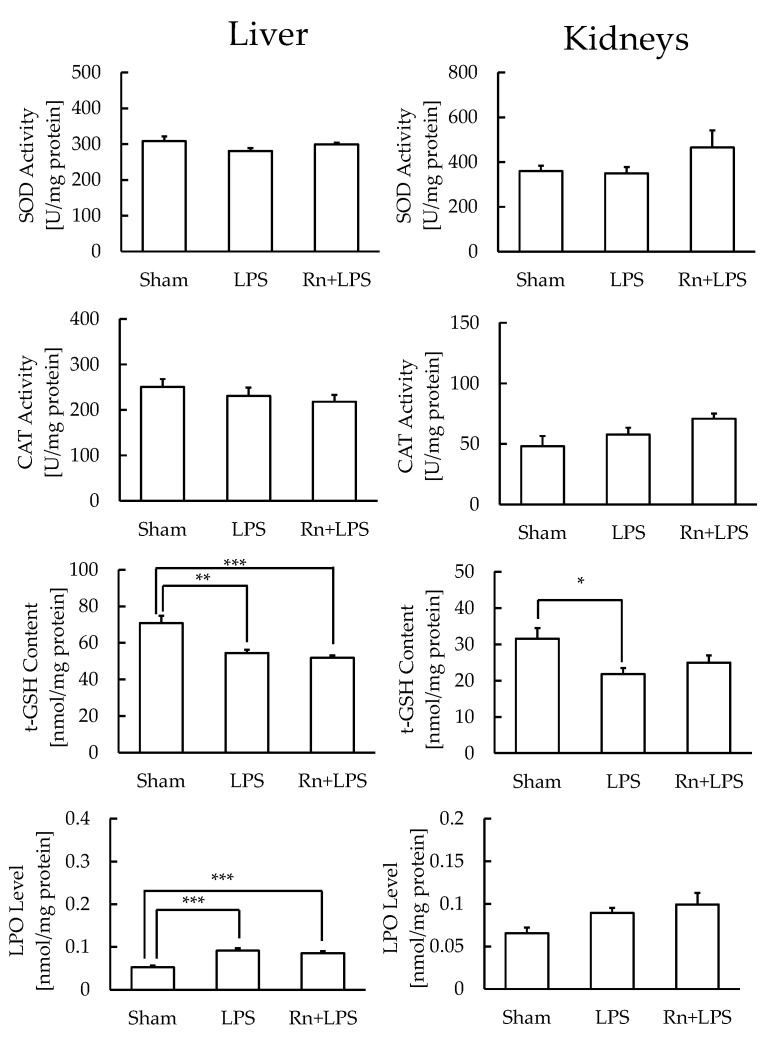
Effect of radon (Rn) inhalation on oxidative stress associated with LPS administration in the liver and kidneys. The results are presented as the mean ± SEM. N = 7; * *p* < 0.05, ** *p* < 0.01, *** *p* < 0.001; all experiments were conducted once; * *p* < 0.05, ** *p* < 0.01, *** *p* < 0.001; sham, mouse group without any treatment; sham + LPS, mouse group that inhaled sham followed by LPS (1 mg/kg body weight); Rn + LPS, mouse group that inhaled radon (2000 Bq/m^3^) followed by LPS (1 mg/kg body weight); SOD, superoxide dismutase; CAT, catalase; GSH, glutathione; LPO, lipid peroxide.

**Figure 4 ijerph-19-10632-f004:**
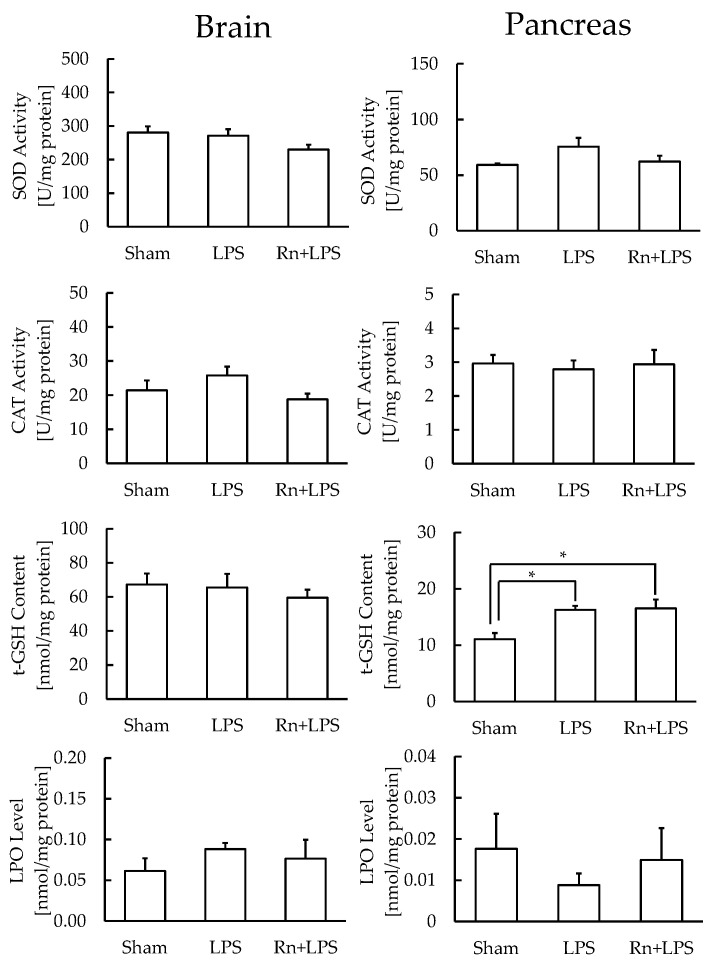
Effect of radon (Rn) inhalation on oxidative stress associated with lipopolysaccharide (LPS) administration in the brain and pancreas. The results are presented as the mean ± SEM. N = 7; * *p* < 0.05; all experiments were conducted once; * *p* < 0.05; sham, mouse group without any treatment; sham + LPS, mouse group that inhaled sham followed by LPS (1 mg/kg body weight); Rn + LPS, mouse group that inhaled radon (2000 Bq/m^3^) followed by LPS (1 mg/kg body weight); SOD, superoxide dismutase; CAT, catalase; GSH, glutathione; LPO, lipid peroxide.

**Figure 5 ijerph-19-10632-f005:**
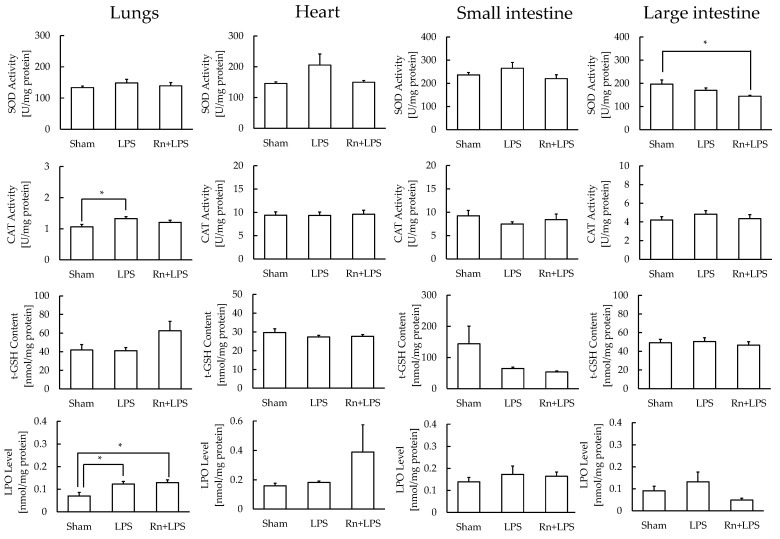
Effect of radon (Rn) inhalation on oxidative stress associated with lipopolysaccharide (LPS) administration in the lungs, heart, small intestine, and large intestine. The results are presented as the mean ± SEM. N = 6–7; * *p* < 0.05; all experiments were conducted once; * *p* < 0.05; sham, mouse group without any treatment; sham + LPS, mouse group that inhaled sham followed by LPS (1 mg/kg body weight); Rn + LPS, mouse group that inhaled radon (2000 Bq/m^3^) followed by LPS (1 mg/kg body weight); SOD, superoxide dismutase; CAT, catalase; GSH, glutathione; LPO, lipid peroxide.

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
