# Peer review of "Immunomodulatory Effects of Radon Inhalation on Lipopolysaccharide-Induced Inflammation in Mice"

_ijerph, 2022, doi:10.3390/ijerph191710632_

Round 1

Reviewer 1 Report

The manuscript's authors: “Basic study on the enhancing effect of radon inhalation on immunomodulatory function in LPS-induced inflammation” show that radon may have immunomodulatory effects on LPS-induced inflammation in vivo. The study may be important to immunologists and environmental toxicologists as well as scientists with interests in oxidative stress and immunology. The study found differences in the levels of anti-oxidative stress markers as well as the concentrations of cytokines in Radon followed by LPS exposure and LPS exposure alone and in control animals (without any exposure).  

Overall, the data show that radon has anti-inflammatory effects in vivo. While the data are novel, the manuscript could have been in better shape if the following concerns are addressed:

1.    Several details are missing in the material and methods section. For example, how and where radon exposure was performed. The n number of mice used in the experiments and how many times the experiments were repeated. The reference (catalog number) of the Bio-plex assay kit is not provided. The references (catalog numbers) of the anti-oxidants assay kits are also missing. Details for the collection and preservation of tissues used to measure anti-oxidant responses are missing. Were tissues flash frozen in liquid nitrogen without any reagent or were they collected in a special buffer and then frozen for further analysis? The authors need to inform the reader of their methodology.

2.    Not clear whether mice were perfused before collecting the tissues. Collecting tissues with peripheral in them would suggest that the authors are still looking at what might happen in the bloodstream and not necessarily specific to the tissues. Taking the time to perfuse the entire mouse would completely remove peripheral blood in every tissue including the brain. Is this relevant to your study?

3.    Anti-oxidants alone are measured, pro-oxidants are not measured. Oxidative stress is an imbalance between pro-oxidants and anti-oxidants. Why are the authors looking only at anti-oxidants? In tissues, for example, 4-Hydroxynonenal is a marker for oxidative stress and Malondialdehyde (MDA) is the most commonly used marker for oxidative stress.

4.    The authors mentioned tissue damage due to oxidative stress without looking at markers of tissue damage.

5.    The authors indicated that radon inhalation significantly decreased IL-6 in comparison to LPS alone (Line 114) but the experiment was only done once.

6.    It would be very helpful to add the detection limit of every cytokine measured with this kit. Concentrations of cytokines in Figure 2 are pretty low.

7.    The figure legends are not detailed enough to guide the reader. Authors should add whether the concentrations of cytokines are from the plasma or from tissue supernatant.

8.    Without using flow cytometry and intracellular cytokine staining, we would not know whether the cytokines measured are from Th1 cells, Th2 cells, or Treg cells. Epithelial and endothelial cells release cytokines. The source of the cytokines is not addressed in the experiments herein. I would use the world Th1 prone, Th2 prone, or Treg prone cytokines or another world.

9.    Why are the authors not using multicolor flow cytometry to see where these cytokines are coming from?

10. Again, the limit of detection for LPO is missing.

11. In vitro studies could have been helpful. The authors could sort T cells and expose them to radon before stimulating them with one concentration of LPS and collecting supernatants at different time points. Also, total protein can be extracted to assess molecular pathways that may be activated/inhibited in the presence or absence of radon.

12. The title could be simpler. For example Immunomodulatory effects of radon inhalation in LPS-induced inflammation. You don’t have to use this title, it's up to you but I think the simpler and easier, the better would be your paper. 

Author Response

Reviewer 1

The manuscript's authors: "Basic study on the enhancing effect of radon inhalation on immunomodulatory function in LPS-induced inflammation" show that radon may have immunomodulatory effects on LPS-induced inflammation in vivo. The study may be important to immunologists and environmental toxicologists as well as scientists with interests in oxidative stress and immunology. The study found differences in the levels of anti-oxidative stress markers as well as the concentrations of cytokines in Radon followed by LPS exposure and LPS exposure alone and in control animals (without any exposure). 

Overall, the data show that radon has anti-inflammatory effects in vivo. While the data are novel, the manuscript could have been in better shape if the following concerns are addressed:

Response: Dear Reviewer, thank you for your efforts and time in evaluating our manuscript. We appreciate your critical comments and suggestions. We have revised the manuscript following your suggestions and provided point-by-point responses to each comment.

  1. Several details are missing in the material and methods section. For example, how and where radon exposure was performed.

Response: Thank you for highlighting this. Accordingly, we have revised our manuscript and added a new subsection, "Experimental Design," to the Materials and methods section and updated the "Radon inhalation" section.

2.2 Experimental design

The mice were divided into three groups (n = 7, in each group): no treatment (sham, control), LPS administration after sham inhalation (LPS), and LPS administration after radon inhalation (Rn + LPS). The mice in the sham group received no treatment, and those in the LPS group received a sham inhalation followed by intraperitoneal administration of LPS at 1.0 mg/kg for 4 h. The mice in Rn + LPS group inhaled radon at 2000 Bq/m3 for 24 h, followed by administration of LPS at 1.0 mg/kg. After 4 h of LPS administration, the mice were euthanized using CO2. The experiments were performed once.

LPS administration induces oxidative stress [15], and radon inhalation activates an-tioxidative functions [5]. Therefore, we also studied the effects of radon therapy on the antioxidative functions of different organs of the mice by assessing the levels of LPO and other antioxidant enzymes. The dose and gap between LPS administration and euthana-sia were determined based on our preliminary study, in which we tested the effects of different doses of LPS on inflammatory responses based on a previous study [15]. LPS administration at 0.4 mg/kg was sufficient to induce inflammatory responses (Figure A1). However, after 24 h of LPS administration, the levels of interferon (IFN) and IL-10 were almost equivalent to those in control. Considering these results and those of pre-vious studies [15], we chose a dose of 1.0 mg/kg LPS for 4 h.  (line 77–94)

2.3 Radon inhalation: Radon inhalation was performed using a radon exposure system, which we have developed and reported in our previous study [7]. DOLL STONEs (Nigyotoge gen-shiryoku sangyo Co., Ltd. Tomata-gun, Okayama, Japan) were used as radon sources. The DOLL STONEs were set in a desiccator, and then radon in the desiccator was trans-ferred to each mouse cage using an air pump. (lines 100–104)

The n number of mice used in the experiments and how many times the experiments were repeated.

Response: Thank you for highlighting this. We used seven mice in each group (line 78), and the experiments were performed once (line 84).

The reference (catalog number) of the Bio-plex assay kit is not provided.

Response: We apologize for missing this information. We have updated the catalog numbers of the kits and updated the Cytokines assay section as follows:

“The levels of cytokines IL-1β, -2, -6, -10, -12(p70), IFN-γ, and TNF-α were detected using the Bio-Plex Pro Mouse Cytokine Th1 Plex Panel kit (catalog no. L6000004C6; Bio-Rad, Hercules, CA, USA, Bio-Plex Pro Mouse Cytokine Th1 Plex Panel, 1 x 96-well, includes coupled magnetic beads, detection antibodies, standards, assay buffer, wash buffer, detection antibody diluent, streptavidin-PE, flat bottom plate, sealing tape, standard diluent, sample diluent, for the detection of IL-1β, IL-2, IL-6, IL-10, IL-12 (p70), IFN-γ, TFN-α). IL-4 and GM-CSF were determined using the Bio-Plex Pro Mouse Cyto-kine Singleplex Set kits (catalog no. 171G5005M and 171G5016M, respectively; Bio-Rad).” (lines 120-127)

The references (catalog numbers) of the anti-oxidants assay kits are also missing.

Response: We apologize for missing this information. We have updated the catalog numbers of the kits and updated the catalog numbers of the kits used for antioxidants assay. The catalog numbers of the kits used for SOD, CAT, t-GSH, LPO, and total protein were S311 (Dojindo Molecular Technologies, Inc., Kumamoto, Japan), PQ01 (Cayman Chemical, Ann Arbor), 707002 (OXIS Health Products, Inc., Portland, OR, USA), 21023 (OXIS Health Products, Inc.), and 21012 (Dojindo Molecular Technologies, Inc., Kumamoto, Japan), respectively. (lines 138, 145, 153, 159, and 170)

Details for the collection and preservation of tissues used to measure anti-oxidant responses are missing. Were tissues flash frozen in liquid nitrogen without any reagent or were they collected in a special buffer and then frozen for further analysis? The authors need to inform the reader of their methodology.

Response: Apologies for the confusion. Following your suggestion, we have revised the “Sample Preparation” section as follows:

“2.5 Sample preparation: Blood was collected from the heart of all mice immediately after euthanasia using CO2 and centrifuged at 3000 × g for 5 min at 4 °C. The upper aqueous layer was collected and preserved at −80 °C until their use for cytokine assays. The brain, lungs, heart, liver, pancreas, kidneys, and small and large intestines were quickly removed without perfu-sion. Then, the organs were washed using distilled water before storing them at −80 °C until further analysis.” (lines 112–118).

  1. Not clear whether mice were perfused before collecting the tissues. Collecting tissues with peripheral in them would suggest that the authors are still looking at what might happen in the bloodstream and not necessarily specific to the tissues. Taking the time to perfuse the entire mouse would completely remove peripheral blood in every tissue including the brain. Is this relevant to your study?

Response: Your comment is very helpful in clarifying the radon effects. Mice were not perfused in this study. Peripheral blood may affect the results of this study. However, there are no reports on whether peripheral blood affects the changes of cytokines and antioxidant substances following radon inhalation. However, as our study aimed to decipher the effects of radon on LPS-induced inflammation, addressing this aspect is beyond the scope of the study. Nevertheless, we will clarify it in the future.

  1. Anti-oxidants alone are measured, pro-oxidants are not measured. Oxidative stress is an imbalance between pro-oxidants and anti-oxidants. Why are the authors looking only at anti-oxidants? In tissues, for example, 4-Hydroxynonenal is a marker for oxidative stress and Malondialdehyde (MDA) is the most commonly used marker for oxidative stress.

Response: We apologize for the confusion. However, in this study, we measured the levels of lipid peroxides (LPO), a major pro-oxidant. Several studies have shown that free radicals and LPO play an important role in the oxidative stress balance between prooxidant and antioxidant activities. The LPO assay kit we used in this study can assay the total of 4-hydroxynonenal and malondialdehyde. MDA is easily detected in organs and used to measure oxidative stress. To avoid confusion, we revised the text as follows:

“The LPO assay is based on the reaction of the chromogenic reagent, N-methyl-2-phenylidole, with malondialdehyde, a product of lipid peroxidation, and 4-hydroxyalkenals at 45 °C.” (line 163–165)

  1. The authors mentioned tissue damage due to oxidative stress without looking at markers of tissue damage.

Response: LPS administration induces damage in the brain [24, 25], lungs [26], heart [15], liver [27], pancreas [28], kidneys [29, 30], and intestines [31]. Thus, oxidative stress and inflammation may play important roles in the development of LPS-induced damage. (line 273–275) As described in response to your previous question, in this study, we assessed oxidative stress marker LPO, which shows the total of 4-hydroxynonenal and malondialdehyde. 

  1. The authors indicated that radon inhalation significantly decreased IL-6 in comparison to LPS alone (Line 114) but the experiment was only done once.

Response: We agree that the absence of replications may be a limitation of our study. However, because of the requirement of using a minimal number of animals to adhere to the rules of the ethical review board, we could not repeat the experiments.

  1. It would be very helpful to add the detection limit of every cytokine measured with this kit. Concentrations of cytokines in Figure 2 are pretty low. 

Response: Data were acquired using Bio-Plex 200 (Bio-Rad). The limits of detections of IL-1β, IL-2, IL-4, IL-6, IL-10, IL-12(p70), IFN-γ, GM-CSF, TNF-α, IL-5, IL-9, IL-13, and IL-17A were 9.4, 0.6, 2.1, 0.2, 1.0, 2.3, 1.2, 5.6, 1.4, 0.3, 12.5 38.7, and 0.8 pg/mL, respectively. (lines 130–132)

  1. The figure legends are not detailed enough to guide the reader. Authors should add whether the concentrations of cytokines are from the plasma or from tissue supernatant.

Response: Apologies for the confusion. We revised the figure legends of each figure and added “in serum” to the legends of Figures 1 and 2. (Please see the Figure legends).

  1. Without using flow cytometry and intracellular cytokine staining, we would not know whether the cytokines measured are from Th1 cells, Th2 cells, or Treg cells. Epithelial and endothelial cells release cytokines. The source of the cytokines is not addressed in the experiments herein. I would use the world Th1 prone, Th2 prone, or Treg prone cytokines or another world.

Response: We revised them (line 21, 57, 58, 177, 155, 192, 200, 198, 201, 203, 254, 261, 402) This comment is useful for future study. We will try to clarify it soon.

  1. Why are the authors not using multicolor flow cytometry to see where these cytokines are coming from?

Response: We agree that using multicolor flow cytometry is critical to clarify the mechanisms of radon therapy. However, we lack appropriate expertise. We will address your suggestion in our future study by collaborating with the specialists. Thank you for your insightful suggestion. (lines 306–307)

  1. Again, the limit of detection for LPO is missing.

Response: The LPO levels were divided by protein levels. Therefore, the levels seem to be low. The limit of detection was 0.1 nmol/mL final concentration, corresponding to an absorbance value of approximately 0.011. Each absorbance obtained by the assays is more than 0.011. We have added the following information to the text:

“The limitation of detection of LPO level was 0.1 nmol/mL, with an absorbance value of approximately 0.011.”(lines 166–168)

  1. In vitro studies could have been helpful. The authors could sort T cells and expose them to radon before stimulating them with one concentration of LPS and collecting supernatants at different time points. Also, total protein can be extracted to assess molecular pathways that may be activated/inhibited in the presence or absence of radon.

Response: We completely agree with your comment. The problem is that radon is hardly dissolved in the water. In addition, α-ray emitted from radon can travel only several micrometers in the water. Therefore, most α-rays may not hit cells. However, we are trying to construct an in vitro experimental system to evaluate radon effects. Therefore, we believe we can overcome these limitations in our future studies. We appreciate your suggestion.

  1. The title could be simpler. For example Immunomodulatory effects of radon inhalation in LPS-induced inflammation. You don't have to use this title, it's up to you but I think the simpler and easier, the better would be your paper.

Response: Thank you for your suggestion. We aggreed to your suggestion and revised the title to "Immunomodulatory effects of radon inhalation on lipopolysaccharide-induced inflammation in mice". (Title)

Reviewer 2 Report

The manuscript entitled “Basic study on the enhancing effect of radon inhalation on immunomodulatory function in LPS-induced inflammation” addresses the beneficial effect of radon inhalation in vivo in lipopolysaccharide-triggered inflammation in mice. The authors also explored the associated molecular mechanisms. Radon inhalation demonstrated favorable anti-inflammatory effects that were established by dampening the production of Th1- and Th17- related cytokines. However, radon did not display antioxidant effects against LPS-triggered organ oxidative stress.

Comments:     

1) In the experimental design of the animal study (section 2.), why did not the authors incorporate an additional experimental group (control + radon). This group may reveal any potential toxicity of the test agent radon inhalation in mice at the indicated dose.

2) What is the LD50 for radon in mice? Is the used dose safe?

3) The Material and methods section is not adequately described. Elaboration of all the used methodology and addition of all the used kits and reagents along with the catalog number is needed.

- A detailed description of cytokine assays is essential and the authors are advised to add the cat no. for all the ELISA kits along with the cut-off values ..etc.

- Likewise, a detailed description of the antioxidant assays is essential, and the authors are advised to add the cat no. for all the kits and add a proper citation to each assay. The collective description of all the antioxidant assays might not be adequate.

4) Please make a separate section for the explanation of the experimental design with enough details about the number of used animals used in each experimental group.

- Detailed explanation is needed to clearly tell how many times was radon administered for mice in LPS and LPS +Rn gps? Was it twice?

- Was LPS administered just once? What was the vehicle for LPS administration? Did the control animals receive the vehicle only?

- When were the animals euthanized? After the LPS by 4 hours? Why did not the authors try a longer time-point? This should be clearly explained in the material and methods section.

5) Since no significant changes were observed in the antioxidant parameters in organs despite the previous literature that LPS depletes organ antioxidant capacity/ markers, would that imply that the used dose (1 mg/kg, i.p.) was not enough to elicit reliable induction of oxidative stress? Or whether the authors should have tried a longer time point for animal euthanization?

6) In the statistical analysis section, did the authors check data normality and homogeneity before proceeding to one-way ANOVA?

7) In the experimental design section, how did the authors decide on the dose of the radon inhalation in mice? How is the dose relevant to the human dose using the Human effective dose (HED) formula= animal dose x animal Km/ human Km (Nair AB, Jacob S. A simple practice guide for dose conversion between animals and humans. J Basic Clin Pharm. 2016 Mar;7(2):27-31). Authors are advised to address this point and add the answers to the comment in section 2.1 (experimental design). Please also provide proper citations for selecting such doses.

8) In the experimental design section, how did the authors decide on the dose of the LPS in mice? Authors are advised to address this point and add the answers to the comment in section 2. 1 (experimental design). Please also provide proper citations for selecting such a dose.

9) The authors are advised to show the effects of radon inhalation on LPS-triggered inflammation with a specific focus on the histology changes in different organs. This would corroborate the current findings.  

10) The authors are also advised to show the effects of radon inhalation on LPS-triggered inflammation with a specific focus on using advanced techniques such as western blotting or immunohistochemistry. This would corroborate the current findings.

11) In figure 1, it is essential to avoid confusion of readers. Hence, the authors are advised to clearly state where the measurement of the 3 cytokines was done. Was it in serum or which organ? Please add this piece of data to the title of section 3.1.

-        Please address this issue in the entire manuscript.

12) To make all figure legends stand-alone, authors are advised to add the full name of the used abbreviations at the end of each legend.

13) In figure 3, the authors have investigated the levels of antioxidant markers in several organs including the liver, kidney, brain, and pancreas. Why did not the authors investigate the levels of cytokines in the 4 organs? This would give better insight to the data of the current study.

14) Likewise, why did not the authors investigate the levels of the antioxidant markers in serum or plasma as done in figures 1 and 2 for the cytokines? 

Author Response

Reviewer 2

The manuscript entitled "Basic study on the enhancing effect of radon inhalation on immunomodulatory function in LPS-induced inflammation" addresses the beneficial effect of radon inhalation in vivo in lipopolysaccharide-triggered inflammation in mice. The authors also explored the associated molecular mechanisms. Radon inhalation demonstrated favorable anti-inflammatory effects that were established by dampening the production of Th1- and Th17- related cytokines. However, radon did not display antioxidant effects against LPS-triggered organ oxidative stress.

Response: Dear Reviewer, thank you for your efforts and time in evaluating our manuscript. We appreciate your critical comments and suggestions. We have revised the manuscript following your suggestions and provided point-by-point responses to each comment.

Comments:     

 1) In the experimental design of the animal study (section 2.), why did not the authors incorporate an additional experimental group (control + radon). This group may reveal any potential toxicity of the test agent radon inhalation in mice at the indicated dose.

 Response: We have already reported that radon inhalation activates antioxidative functions [5] and the changes in cytokines [11]. These previous studies suggest no potential toxicity of radon inhalation. Furthermore, reducing the number of mice was mandatory from an ethical point of view. Therefore, we preferred not to put up an additional group.

2) What is the LD50 for radon in mice? Is the used dose safe?

Response: As far as we know, there are no reports of the LD50 for radon in mice. Irradiation from radon, an α-emitter, is completely different from X- or γ-ray. Since radon is a gas, it enters the body through the lungs. When humans are exposed to radon under this experimental condition, it is estimated that the effective dose is about 0.2 mSv. Therefore, the dose in this study is considered to be safe.

3) The Material and methods section are not adequately described. Elaboration of all the used methodology and addition of all the used kits and reagents along with the catalog number is needed.

Response: Thank you for your suggestion. We have revised the entire Materials and Methods section with a few new sections. In the revised Materials and Methods section, we have provided more details about the experimental design and radon inhalation. We also updated the sample preparation and cytokine and antioxidant assay section. The details of the kits used in this study have been updated in respective sections as appropriate. (Please see the revised Materials and Methods section; lines 77–171)

- A detailed description of cytokine assays is essential and the authors are advised to add the cat no. for all the ELISA kits along with the cut-off values etc.

Response: Thank you for your suggestion. We have revised the “Cytokine assay” section as follows:

“2.6 Cytokines assay: The levels of cytokines IL-1β, -2, -6, -10, -12(p70), IFN-γ, and TNF-α were detected using the Bio-Plex Pro Mouse Cytokine Th1 7-Plex Panel kit (catalog no. L6000004C6; Bio-Rad, Hercules, CA, USA, Bio-Plex Pro Mouse Cytokine Th1 Plex Panel, 1 x 96-well, includes coupled magnetic beads, detection antibodies, standards, assay buffer, wash buffer, detection antibody diluent, streptavidin-PE, flat bottom plate, sealing tape, standard diluent, sample diluent, for the detection of IL-1β, IL-2, IL-6, IL-10, IL-12 (p70), IFN-γ, TFN-α). IL-4 and GM-CSF were determined using the Bio-Plex Pro Mouse Cytokine Singleplex Set kits (catalog no. 171G5005M and 171G5016M, respectively; Bio-Rad). The assay was consigned to Okayama University Hospital Biobank.. Data were acquired using a Bio-Plex 200 system (Bio-Rad). The limits of detections of IL-1β, IL-2, IL-4, IL-6, IL-10, IL-12(p70), IFN-γ, GM-CSF, TNF-α, IL-5, IL-9, IL-13, and IL-17A were 9.4, 0.6, 2.1, 0.2, 1.0, 2.3, 1.2, 5.6, 1.4, 0.3, 12.5 38.7, and 0.8 pg/mL, respectively. (lines 119–132)

- Likewise, a detailed description of the antioxidant assays is essential, and the authors are advised to add the cat no. for all the kits and add a proper citation to each assay. The collective description of all the antioxidant assays might not be adequate.

Response: The methods of assays are completely the same as we reported in previous studies. We wanted to reduce plagiarism; therefore, we avoided their full description in our original manuscript. However, we added a detailed description of the antioxidant assays used here and revised the contents as follows:

“2.7 Antioxidant assay: The levels of SOD, CAT, t-GSH, LPO, and total protein in the organs were assessed using individual assay kits following previously described protocols [6,16]. For the SOD and glutathione assays, samples were homogenized on ice in a 10 mM phosphate buff-ered solution (PBS; pH 7.4), and homogenates were used for further analysis. SOD activ-ity was determined using the SOD Assay Kit-WST (catalog no. S311; Dojindo Molecular Technologies, Inc., Kumamoto, Japan) based on the nitroblue tetrazolium (NBT) reduc-tion method [17]. Briefly, the homogenates were centrifuged at 12,000 × g for 45 min at 4 °C, and the supernatants were collected. SOD activity in the supernatants was meas-ured by the extent of inhibition of NBT reduction measured at 450 nm using a spectro-photometer (Viento XS; DS Pharma Biomedical Co., Ltd, Osaka, Japan). Percentage inhi-bition of one-unit enzyme activity was defined as 50% inhibition of NBT reduction.

CAT activity was measured using a Catalase Assay Kit (catalog no. PQ1; Cayman Chemical, Ann Arbor), which uses a method based on the reaction of the enzyme with methanol in the presence of an optimal concentration of H2O2. The formaldehyde pro-duced was measured colorimetrically with 4-amino-3-hydrazino-5-mercapto-1,2,4-triazole (Purpald) as the chromogen. Purpald specifically forms a bicyclic heterocycle with aldehydes, which changes from colorless to a purple color upon oxidation [18,19]. The absorbance was read at 540 nm using a plate reader (Viento XS).

The t-GSH content was measured using the Bioxytech GSH-420TM assay kit (catalog no. 707002; OXIS Health Products, Inc., Portland, OR, USA). This assay is based on the formation of a chromophoric thione, which is directly proportional to the t-GSH concen-tration and can be measured at 420 nm. Briefly, sample tissue homogenates were mixed with an ice-cold 10 mM PBS and 7.5% trichloroacetic acid solution. The homogenates were centrifuged at 3,000 × g for 10 min at 4 °C, and the supernatants were collected to measure t-GSH.

LPO level was estimated using the Bioxytech LPO-586TM assay kit (catalog no. 21023; OXIS Health Products, Inc.). Briefly, the samples were placed in 10 mM PBS (pH 7.4), and 10 μL 0.5 M butylated hydroxytoluene in acetonitrile was added per mL of the buffer-tissue mixture. The tissue was homogenized, the homogenate was centrifuged at 15,000 × g for 10 min at 4 °C, and the supernatant was used for the biochemical assay. The LPO assay is based on the reaction of the chromogenic reagent, N-methyl-2-phenylidole, with malondialdehyde, a product of lipid peroxidation, and 4-hydroxyalkenals at 45 °C. The optical density of the colored product was read at 586 nm using a spectrophotometer. The limitation of detection of LPO level was 0.1 nmol/mL, with an absorbance value of approximately 0.011.

The protein content of each sample was measured by the Bradford method using a Protein Quantification Kit-Rapid (catalog no. 21012; Dojindo Molecular Technologies, Inc.) [20]. ” (lines 133–171)

  1. Baehner, R. L.; Murrmann, S. K.; Davis, J.; Johnston, R. B. Jr. The role of superoxide anion and hydrogen peroxide in phagocytosis-associated oxidative metabolic reactions. J. Clin. Invest. 1975, 56, 571-576.
  2. Johansson, L. H.; Borg, L. A. H. A spectrophotometric method for determination of catalase activity in small tissue samples. Anal. Biochem. 1988, 174, 331–336.
  3. Wheeler, C. R.; Salzman, J. A.; Elsayed, N. M.; Omaye, S. T.; Korte, D. W. Jr. Automated assays for superoxide dismutase, catalase, glutathione peroxidase, and glutathione reductase activity. Anal. Biochem. 1990, 184, 193–199.
  4. Bradford, M. M. A rapid and sensitive method for the quantitation of microgram quantities of protein utilizing the principle of protein-dye binding. Anal. Biochemi. 1976, 72, 248-254.

4) Please make a separate section for the explanation of the experimental design with enough details about the number of used animals used in each experimental group.

Response: Thank you for your suggestion. As explained earlier, we have added a new section "Experimental design” to the revised manuscript as follows:

“2.2 Experimental design: The mice were divided into three groups (n = 7, in each group): no treatment (sham, control), LPS administration after sham inhalation (LPS), and LPS administration after radon inhalation (Rn + LPS). The mice in the sham group received no treatment, and those in the LPS group received a sham inhalation followed by intraperitoneal administration of LPS at 1.0 mg/kg for 4 h. The mice in Rn + LPS group inhaled radon at 2000 Bq/m3 for 24 h, followed by administration of LPS at 1.0 mg/kg. After 4 h of LPS administration, the mice were euthanized using CO2. The experiments were performed once.

LPS administration induces oxidative stress [15], and radon inhalation activates an-tioxidative functions [5]. Therefore, we also studied the effects of radon therapy on the antioxidative functions of different organs of the mice by assessing the levels of LPO and other antioxidant enzymes. The dose and gap between LPS administration and euthana-sia were determined based on our preliminary study, in which we tested the effects of different doses of LPS on inflammatory responses based on a previous study [15]. LPS administration at 0.4 mg/kg was sufficient to induce inflammatory responses (Figure A1). However, after 24 h of LPS administration, the levels of interferon (IFN) and IL-10 were almost equivalent to those in control. Considering these results and those of pre-vious studies [15], we chose a dose of 1.0 mg/kg LPS for 4 h. Furthermore, in the prelim-inary study, the levels of IL-1β, -2, -4, -5, -6, -9, -10, -12(p70), -13, -17, IFN-γ, granulo-cyte-macrophage colony-stimulating factor (GM-CSF), and TNF-α were assessed (Figure A1). However, the levels of IL-5, IL-9, IL-13, and IL-17 levels remain unchanged follow-ing radon inhalation or LPS administration. Therefore, we did not assess these cytokines in the present study." (lines 77–97)

- Detailed explanation is needed to clearly tell how many times was radon administered for mice in LPS and LPS +Rn gps? Was it twice?

Response: The mice in Rn + LPS group inhaled radon at 2000 Bq/m3 for 24 h, followed by admin-istration of LPS at 1.0 mg/kg. (line 82–83). It was performed once.

- Was LPS administered just once? What was the vehicle for LPS administration? Did the control animals receive the vehicle only?

Response: The mice were intraperitoneally administered LPS (lipopolysaccharide from Escherichia coli O127:B8 Sigma-Aldrich, Tokyo, Japan) at a dose of 1.0 mg/kg body weight. Mice in the sham group were not administered anything. (line 109)

- When were the animals euthanized? After the LPS by 4 hours? Why did not the authors try a longer time-point? This should be clearly explained in the material and methods section.

Response: At 4 h after LPS administration, the mice were euthanized using CO2. (line 83–84) The dose and gap between LPS administration and euthanasia were determined based on our preliminary study, in which we tested the effects of different doses of LPS on in-flammatory responses based on a previous study [15]. LPS administration at 0.4 mg/kg was sufficient to induce inflammatory responses (Figure A1). However, after 24 h of LPS administration, the levels of interferon (IFN) and IL-10 were almost equivalent to those in control. Considering these results and those of previous studies [15], we chose a dose of 1.0 mg/kg LPS for 4 h. We have updated these in the text. (lines 88–94)

5) Since no significant changes were observed in the antioxidant parameters in organs despite the previous literature that LPS depletes organ antioxidant capacity/ markers, would that imply that the used dose (1 mg/kg, i.p.) was not enough to elicit reliable induction of oxidative stress? Or whether the authors should have tried a longer time point for animal euthanization?

Response: The used dose may not be enough to elicit reliable induction of oxidative stress. Moreover, we may confirm oxidative stress with a longer time for animal euthanization. However, as observed in our preliminary study, some cytokines may return to their normal level, which could have affected our conclusions. Therefore, we choose a 4 h gap for euthanization.

6) In the statistical analysis section, did the authors check data normality and homogeneity before proceeding to one-way ANOVA?

Response: Yes, we checked them.

7) In the experimental design section, how did the authors decide on the dose of the radon inhalation in mice? How is the dose relevant to the human dose using the Human effective dose (HED) formula= animal dose x animal Km/ human Km (Nair AB, Jacob S. A simple practice guide for dose conversion between animals and humans. J Basic Clin Pharm. 2016 Mar;7(2):27-31). Authors are advised to address this point and add the answers to the comment in section 2.1 (experimental design). Please also provide proper citations for selecting such doses.

Response: Radon concentration was decided based on that used for radon therapy. HED seemed to be used to estimate the start dose for clinical trials of medicines. Therefore, it may not be appropriate to estimate the absorbed doses for mice.

8) In the experimental design section, how did the authors decide on the dose of the LPS in mice? Authors are advised to address this point and add the answers to the comment in section 2. 1 (experimental design). Please also provide proper citations for selecting such a dose.

Response: We decided on the experimental condition by conducting a preliminary study (Figure A1). As explained in our earlier response, we have revised the Experimental design section and updated the related information. Please see lines 88–94.

9) The authors are advised to show the effects of radon inhalation on LPS-triggered inflammation with a specific focus on the histology changes in different organs. This would corroborate the current findings.  

Response: Thank you for this insightful comment. It was not impossible to perform such analysis due to the expiration of reseach project. In this study, we tried to evaluate the whole body effects of LPS-induced damage in each organ. In the future, we will examine the effects of radon inhalation on LPS-triggered inflammation in a specific organ focusing on the histology changes. (lines 307–310)

10) The authors are also advised to show the effects of radon inhalation on LPS-triggered inflammation with a specific focus on using advanced techniques such as western blotting or immunohistochemistry. This would corroborate the current findings.

Response: We appreciate this insightful comment. We agree that western blotting and immunohistochemistry are useful techniques to examine the mechanism of the inhibitory effects of radon inhalation on LPS-triggered inflammation in a specific organ. However, we could not conduct the experiments in this study as we focused on the effects of radon inhalation on alleviating LPS-induced damage in each organ. We will use the techniques when we examine the effects of radon inhalation on LPS-triggered inflammation in a specific organ. (lines 307–310)

11) In figure 1, it is essential to avoid confusion of readers. Hence, the authors are advised to clearly state where the measurement of the 3 cytokines was done. Was it in serum or which organ? Please add this piece of data to the title of section 3.1.

-        Please address this issue in the entire manuscript.

Response: We added "in serum" in the title of section 3.1 and respective Figure legends, as appropriate (lines 177, 186, 193, 200, and 401).

12) To make all figure legends stand-alone, authors are advised to add the full name of the used abbreviations at the end of each legend.

Response: Thank you for the suggestion. We have defined all abbreviations used in Figure in the respective figure legends. Please see the revised Figures. (lines 185, 187-190, 192, 194–198, 216, 218–221, 228, 230–233, 265, 268–271, and 403-406)

13) In figure 3, the authors have investigated the levels of antioxidant markers in several organs including the liver, kidney, brain, and pancreas. Why did not the authors investigate the levels of cytokines in the 4 organs? This would give better insight to the data of the current study.

Response: Thank you for your great advice. I completely agree with it. There was a limitation of our funding. Since we will conduct further research in the future, we will clarify these. 

14) Likewise, why did not the authors investigate the levels of the antioxidant markers in serum or plasma as done in figures 1 and 2 for the cytokines? 

Response: We wanted to assay the antioxidant markers in serum. However, the amount of serum samples was not enough to assay them.

Round 2

Reviewer 1 Report

The authors took care of the comments/concerns that were in the original submission. The paper is now in better shape and easier to read. 

Reviewer 2 Report

The authors have adequately addressed most of the comments